# Research Progress and Prospect of Alfalfa Resistance to Pathogens and Pests

**DOI:** 10.3390/plants11152008

**Published:** 2022-08-01

**Authors:** Bo Yang, Yao Zhao, Zhenfei Guo

**Affiliations:** 1College of Grassland Science, Nanjing Agricultural University, Nanjing 210095, China; yangb@njau.edu.cn; 2Department of Plant Pathology, College of Plant Protection, Nanjing Agricultural University, Nanjing 210095, China; yaozhao@njau.edu.cn

**Keywords:** plant innate immunity, alfalfa, R gene, resistance mechanism, molecular breeding

## Abstract

Alfalfa is one of the most important legume forages in the world and contributes greatly to the improvement of ecosystems, nutrition, and food security. Diseases caused by pathogens and pests severely restrict the production of alfalfa. Breeding resistant varieties is the most economical and effective strategy for the control of alfalfa diseases and pests, and the key to breeding resistant varieties is to identify important resistance genes. Plant innate immunity is the theoretical basis for identifying resistant genes and breeding resistant varieties. In recent years, the framework of plant immunity theory has been gradually formed and improved, and considerable progress has been made in the identification of alfalfa resistance genes and the revelation of the related mechanisms. In this review, we summarize the basic theory of plant immunity and identify alfalfa resistance genes to different pathogens and insects and resistance mechanisms. The current situation, problems, and future prospects of alfalfa resistance research are also discussed. Breeding resistant cultivars with effective resistance genes, together with other novel plant protection technologies, will greatly improve alfalfa production.

## 1. Introduction

Alfalfa (*Medicago sativa*) is the world’s most significant perennial legume forage, with large yields, high digestibility, good palatability, and wide adaptability. Alfalfa is regarded as “the queen of forages” as it contributes greatly to the improvement of ecosystems, nutrition, and food security [1]. In recent years, the need for animal products in China has expanded, and the demand for forage grass has substantially increased. China imported 1.78 million tons of alfalfa hay in 2021 [2,3]. Improving alfalfa yield and quality in China is a critical issue for food supply security.

Pests and diseases are major threats to alfalfa and cause vast economic losses, which not only affect the yield and quality of alfalfa but also cause livestock poisoning [4]. The cultivation of pest-resistant cultivars is the most cost-effective and environmentally friendly way to manage pests and diseases, and it is also a key strategy for ensuring green agriculture’s long-term growth [5]. The identification of pest and disease resistance genes is critical for breeding pest-resistant alfalfa lines, whereas plant immunology provides the theoretical foundation for discovering resistant genes and cultivating resistant plants [6]. Significant advances in plant immunity have been made over the years, and the framework of plant immunity theory has been well established [6]. From the perspective of plant immunity, this paper reviews the identification of resistance genes against pests and diseases (Table 1) and related molecular mechanisms in alfalfa, as well as the discussions on problems for breeding resistant plants and the future of integrated pest and disease management.

## 2. Theory of Plant Immunity: Theoretical Basis for Breeding Resistant Plants to Pathogens and Insects

In the early twentieth century, Biffen discovered the existence of stripe rust resistance genes in wheat [37]. Flor proposed the “gene-to-gene” hypothesis in the 1940s through research on flax and flax rust, laying an important genetic foundation for plant disease resistance research [38]. In 1984, the first avirulent gene, AvrA, was cloned from *Pseudomonas syringae* [39]. In 1992, the first disease resistance gene, HM1, was cloned from maize [40]. Scientists had first defined the fundamental basis of plant immunity theory by 2006, which was basically separated into two layers of the innate immune system [6]. Pattern recognition receptors recognize pathogens, PAMPs (pathogen-associated molecular patterns), or plant-derived molecular patterns, DAMPs (damage-associated molecular patterns), related to infection damage, hence triggering plant PTI (pattern-triggered immunity); the second layer is effector-triggered immunity (ETI), which is initiated by the recognition of pathogen-secreted effectors with plant intracellular immune receptors, NLRs (nucleotide-binding domain and leucine-rich repeat receptors) [6]. PTI is a universal but weak resistance that can prevent the infection of most pathogenic microbes, whereas pathogens evolve effectors into plant cells, blocking the PTI to defeat the plant’s disease resistance. Plants will evolve equivalent NLRs to activate ETI in response to pathogenic effectors, whereas pathogens could escape recognition via effector sequence variation and release novel effectors to interfere with ETI [5]. In the process of evolution, both pathogens and host plants continue to co-evolve in this unending fight, conducting an “arms race” with each other [41]. The latest research shows that the two levels of immune pathways in plants are linked and synergistically trigger the plant immune response, therefore limiting the invasion of pathogenic microbes [42]. Here in this review, we only list some typical plant immune receptor studies as examples, since there are several excellent and comprehensive reviews for the summary of plant immune receptors [43,44].

### 2.1. Plant Resistance Mediated by Pattern Recognition Receptors on Cell Membranes

PAMPs are a class of conserved molecules in pathogens that include bacterial flagellin, elongation factor, peptidoglycan, fungal chitin, chitosan oligosaccharide, oomycete elicitin, and glycosyl hydrolase. Once pathogens attack plants, these PAMPs are recognized by PRR receptors on the plant cell membrane and rapidly stimulate immune responses, such as the burst of reactive oxygen species, the influx of calcium ions, and the accumulation of callose, allowing the plant to resist the invasion of pathogens. As more cell membrane surface receptors have been identified, it is found that they mainly include transmembrane receptor-like kinase (RLK) and receptor-like protein (RLP) [43]. The most commonly reported model receptors include FLS2, Xa21, EFR, CEBiP, etc. [43]. For example, the extracellular domain of FLS2 recognizes bacterial PAMP flagellin, which interacts with the co-receptor BAK1 to form a receptor complex that activates plant immune responses [45]. PsXEG1 secreted by *Phytophthora sojae* triggers plant immunity in tobacco, and the tobacco receptor-like protein NbRXEG1 directly binds and recognizes PsXEG1 using its extracellular domain [46]. Since PsXEG1 homologous proteins are widely present in a variety of pathogenic oomycetes and fungi, engineering plants with NbRXEG1 is a viable strategy for increasing plant broad-spectrum resistance to these pathogens [46,47]. Rice LYP4 and LYP6 are pattern recognition receptors with dual functions, which can recognize bacterial peptidoglycan and fungal chitin and activate plant immune responses against bacteria and fungi [48]. In addition, Chinese scholars have identified Bph3, a gene encoding a receptor-like kinase in rice, which has broad-spectrum and long-lasting resistance to rice planthoppers [49]. Mining and identifying more novel immune receptors will provide valuable genetic resources for improving plant resistance and creating new plant materials with broad-spectrum disease and insect resistance.

### 2.2. Plant Resistance Mediated by Intracellular NLR Immune Receptors

Intracellular immune receptors in plants can directly or indirectly recognize the effectors secreted by pathogens and activate ETI, involving important resistance gene resources for plant disease and insect resistance breeding [6]. Since the first disease resistance gene HM1 was cloned in 1992 from maize, more than 400 disease and insect resistance genes have been cloned from different plants. These genes encode different types of disease and insect resistance proteins, and most of them encode NLR receptor proteins, such as potato NLR receptors Rpivnt1 and R8 against late blight, rice NLR receptors Bph14 and Bph9 against planthoppers, etc. [50,51]. NLR proteins are members of signal transduction ATPases with numerous domains (STAND), including a non-conserved N-terminal domain, a conserved nucleotide-binding (NB) domain, and a C-terminal LRR (leucine-rich repeat) domain [52]. The NB domain in plant NLR is thought to be a molecular switch with ADP- and ATP-bound forms that indicate the “off” and “on” states of NLR signaling [52]. According to their N-terminal domains, NLR receptors can be divided into CC-NB-LRR and TIR-NB-LRR [52,53]. These NLR receptors mainly recognize the effectors of invading organisms through multiple mechanistic models, such as “Direct”, “Decoy”, “Guard”, or “Integrated decoy”, and then specifically initiate downstream immune pathways [52,53]. The number of NLR genes in different plant species ranges from dozens to hundreds, and there is a high degree of diversity between and within species, forming huge NLR groups, indicating that the composition of NLR in the genome of different plant species is evolved independently. Therefore, the genome analysis of plant NLR is important for understanding the evolution of NLRs, different types of NLR proteins, and for discovering new NLR genes [53]. Structural biology has been used to study the Arabidopsis NLR receptor ZAR1, and the results revealed the structural changes of the ZAR1 protein before and after recognizing effectors and proposed the concept of resistosome [54,55]. The latest research shows that different types of NLRs could form a calcium-permeable channel to trigger plant immune signaling [56].

### 2.3. Plant Immune Signaling

Plant immune response is a complex biological process. After the receptor recognizes the corresponding ligand or effector, the immune signal needs to be further transmitted, and finally activate the immune response. This process includes protein phosphorylation/ubiquitination regulation, the synthesis of reactive oxygen species, the activation of the MAPK signaling pathway, the expression of defense response genes, the synthesis of secondary metabolites, etc. [57]. For example, the cytoplasmic receptor-like protein kinase BIK1 binds to pattern recognition receptors and phosphorylates the downstream components, such as the NADPH oxidase RBOHD localized in the cytoplasmic membrane and the kinase MAP3K3/5 of the MAPK cascade pathway, thereby generating reactive oxygen species, activating the MAPK pathway, and then activating the immune response [42,58]. In addition, various families of transcription factors have been reported to be involved in plant immune responses. For example, the Arabidopsis transcription factor WRKY8 regulates the plant immune response to viruses by regulating the expression of key genes in plant abscisic acid and ethylene signaling pathways [59]. Furthermore, a variety of plant hormones are also involved in the plant immune response process. For example, salicylic acid plays a key role in plant defense against biotrophic or hemibiotrophic pathogens, whereas jasmonic acid signaling is required for plant resistance to necrotrophic pathogens or insects [60]. Other hormonal signals, including ethylene and auxin, have also been reported to be involved in plant immune responses [61,62,63].

Annually, about twenty percent of crop yields are lost due to plant diseases and pests, which affect food security worldwide [64]. Pathogens constantly evolve to overcome host resistance. In addition to traditional chemical control strategies, an improved understanding of plant immunity should contribute to the development of new crop protection strategies. Genetic methods to improve crop resistance with plant immune receptors will provide broad-spectrum disease resistance. Although studies on plant immunity are mostly reported on model plants and staple crops, it will be helpful to identify immune receptors and immune-related genes of alfalfa to enable new breeding strategies to enhance alfalfa disease resistance.

## 3. Main Diseases of Alfalfa and its Disease Resistance Mechanisms

### 3.1. Main Diseases and Pathogens of Alfalfa

A variety of pathogens infect alfalfa and cause serious diseases, thus affecting the yield and quality of alfalfa. The pathogens that cause alfalfa disease include oomycetes such as *Phytophthora*, *Pythium*, and *Aphanomyces*; fungi such as *Rhizoctonia*, *Sclerotinia*, rust, powdery mildew, *Verticillium*, *Fusarium*; bacteria such as *Pseudomonas*, *Erwinia*, and *Agrobacterium tumefaciens*; as well as mosaic viruses, nematodes, and parasitic plants [4,65]. The diverse population structure of pathogens results in a high cost of disease management and low disease control efficiency. On the other hand, the co-infection of multiple pathogens in field conditions further raises the difficulties of disease control [66,67].

### 3.2. Bacterial Disease Resistance Genes and Mechanisms in Alfalfa

Many bacteria cause alfalfa diseases, mainly including *Clavibacter insidiosus*, *Ralstonia solanacearum*, *Pseudomonas syringae* pv. *syringae*, *Agrobacterium tumefaciens*, etc [68]. The common symptoms of the bacterial diseases of alfalfa are necrosis and rot, wilting, and tumors. Under high humidity, the lesions develop water-soaked edges and a gel-like bacterial snot. Transcriptomics was utilized to investigate how resistant and susceptible alfalfa cultivars responded to *P. syringae*. Resistance genes such as homologs of rice *Xa21* and soybean *Rpg1-b* were significantly up-regulated when resistant cultivars were infected with *P. syringae* [69]. *MtQRRS1* was identified as a QTL that regulates alfalfa resistance to *R. solanacearum* in *M. truncatula* [7]. The 64 kb region of *MtQRRS1* sites contains 15 candidate genes, seven of which are R genes with typical NLR functional domains, and these genes show sequence polymorphisms in resistant lines, suggesting that variations in these genes regulate *M. truncatula* resistance to *R. solanacearum* [8]. Arabidopsis EFR receptors are pattern recognition receptors on the cell membrane that could recognize the PAMP EF-Tu released by pathogenic bacteria, inducing the PTI response. The transgenic expression of the Arabidopsis EFR receptor in *M. truncatula* significantly enhanced alfalfa resistance to bacterial wilt disease [9]. Interestingly, the transgenic expression of the human lactoferrin-encoding gene in alfalfa dramatically increased the resistance of alfalfa plants to *P. syringae and Clavibacter michiganensis* [10].

### 3.3. Oomycete Disease Resistance Genes and Mechanisms in Alfalfa

*Phytophthora *spp., *Pythium *spp., and *Aphanomyces *spp. are the major oomycete pathogens of alfalfa [70]. These soil-borne pathogens can infect alfalfa seedlings or mature plants, especially in soil fields with high humidity or poor drainage. It is difficult to manage the disease because the oospores of oomycetes could survive in soil or plant residues for several years [70]. There are few reports on the identification of genes associated with *Phytophthora* resistance in alfalfa. The genetic linkage map revealed 3 QTLs in the non-homologous linkage group that explained 6–15% of the variance, suggesting that there are at least three resistance genes, two of which enhance the sensitivity of plants [71]. In addition, the overexpression of *MtERF1-1* in *Medicago truncatula* significantly enhanced the resistance of plants to *P. medicaginis* [11]. The transgenic alfalfa that overexpresses the β-1,3-glucanase gene could significantly reduce the root rot caused by *P. medicaginis* [12].

A QTL at the top of chromosome 3 in *M. truncatula* was involved in the resistance to multiple races of *Aphanomyces euteiches* [20,72,73]. An F-box protein-encoding gene was identified as a candidate resistance gene by a genome-wide association analysis (GWAS) [15]. Interestingly, the sequencing of this gene in 20 resistant or susceptible lines revealed that the nucleotides of the F-box-encoding gene in the resistant lines were mutated, possibly generating a nonfunctional F-box gene. This result suggests that the gene encoding F-box may be a negative regulator of *M. truncatula* resistance to *A. euteiches* [15]. In addition, several secondary metabolites may be involved in the resistance of alfalfa to *A. euteiches*. For example, a transcriptome and proteome analysis found that *A. euteiches* infection significantly induced the up-regulated expression of *M. truncatula* genes in the isoflavone biosynthesis pathway, resulting in the synthesis of the phytoalexin medipterin [74,75]. The expression level of gene-encoding chalcone O-methyltransferase was significantly induced in the resistant line, and 2′-O-methylisoliquiritigenin catalyzed by the enzyme could strongly inhibit mycelium growth and zoospores germination [16]. During infection, the sesquiterpene synthase MtTPS10 of *M. truncatula* was greatly increased, and the sesquiterpenes effectively suppressed the mycelial growth and zoospore germination [17].

### 3.4. Fungal Disease Resistance Genes and Mechanisms in Alfalfa

The common fungal diseases of alfalfa are anthracnose, rust, powdery mildew, *Fusarium* or *Rhizoctonia* root rot, *Verticillium* wilt, etc. These fungi infect the roots or aerial parts of alfalfa throughout the growing season, resulting in significant yield losses [76,77]. The anthracnose of alfalfa, caused by *Colletotrichum trifolii*, is a destructive disease that affects alfalfa worldwide. High temperatures and humidity facilitate the incidence and spread of anthracnose, which is most likely to occur at the end of the summer [78]. When disease-resistant alfalfa plants were infected with *C. trifolii*, the expression of genes involved in the phenylpropanoid metabolic pathway was significantly up-regulated, and more medipterin and sativan were synthesized [79]. The RCT1 gene from *M. truncatula* confers broad-spectrum resistance to anthracnose in alfalfa. RCT1 encoded a disease resistance gene of the TIR-NBS-LRR class, and the expression of *RCT1* in susceptible plants significantly increased plant resistance to anthracnose races 1, 2, and 4 [22]. The *RCT1* gene includes five exons, intron 4 is retained due to alternative splicing, and both normal and alternatively spliced transcripts contribute to resistance [23].

The biotrophic fungus *Erysiphe pisi* causes powdery mildew in *Medicago*. In *M. truncatula*, the powdery mildew resistant genotype Jemalong A17 was hybridized with the susceptible genotype F83005.5. After inoculation with powdery mildew, the hybrid F2 generation showed a 3:1 segregation of the resistant-susceptible phenotype, indicating that there is a single dominant gene in A17 that mediates resistance to powdery mildew, referred to as MtREP1 [25]. Further research revealed that MtREP1 encodes a CC-NB-LRR-type NLR gene. The *MtREP1* gene was transferred into susceptible *M. truncatula*, and the transgenic plants exhibited resistance to powdery mildew [25]. The *MtREP1* gene was shown to be consistently expressed in the resistant genotype Jemalong A17, but not in the susceptible genotype F83005.5. There are no polymorphisms in the two sequences except for a 430-bp deletion in Mtrep1’s 3’UTR region. Further research revealed that epigenetic regulation determines the expression level of the disease resistance gene in susceptible and resistant genotypes, determining plant resistance to powdery mildew [25].

The *Verticillium* wilt of alfalfa is a fungal disease caused by the fungus *Verticillium *spp., which is still a quarantined disease in China. GWAS analysis revealed multiple QTLs related to *Verticillium* wilt resistance on alfalfa chromosomes 1, 7, and 8 [80]. A molecular marker on chromosome 8 has been identified as being important in alfalfa resistance to *Verticillium* wilt [81]. Subsequent research revealed two linked *Verticillium* wilt resistance genes from alfalfa, MsVR38, and MsVR39, both of which belong to the TIR-NBS-LRR family of NLR genes [26]. MsVR39 positively regulated resistance to *Verticillium* wilt, while MsVR38 negatively regulated it [26]. The knockout of *MtVR130*, which is the homologous gene of *MsVR38* in *M. truncatula*, significantly increased the resistance to *Verticillium* wilt compared to the wild type. *M. truncatula* that mutated the *MsVR39* homolog *MtVR140* were more susceptible to *Verticillium* infection [26]. When the highly identical TIR-NBS-LRR pairs are both normally expressed, they are more likely to form heterodimers, preventing the formation of MsVR39 homodimers and increasing alfalfa susceptibility to *Verticillium* wilt [26].

In *M. truncatula*, several important disease resistance-related genes against other fungal pathogens were reported. For example, the overexpression of the ethylene response-related transcription factor *MtERF1-1* or the isoflavone synthase-encoding gene IFS promoted *M. truncatula* resistance to *Rhizoctonia solani* [11,31]. Resistance to *Phoma medicaginis* was dramatically increased in transgenic alfalfa overexpressing isoflavone-O-transmethylase (IOMT) [28]. Furthermore, the heterologous expression of the *M. truncatula* defensin-related gene *MtDef4* in Arabidopsis and wheat greatly enhanced Arabidopsis resistance to downy mildew and wheat tolerance to rust [21,29]. Silencing the *MtABCG10* gene in *M. truncatula* increased its sensitivity to *Fusarium oxysporum* substantially [30].

## 4. Main Alfalfa Insect Pests and Related Resistance Mechanisms

Aphids, weevils, thrips, leaf miners, noctuid moths, white butterflies, grubs, and other alfalfa pests have caused severe losses to alfalfa production. Based on 2017 data, the annual loss of alfalfa in China caused by insect pests exceeded 140 million US dollars [82]. However, few studies have been conducted on the interaction between alfalfa and pests, mainly focusing on *M. truncatula* resistance to aphids. Four resistance genes have been found in *M. truncatula* to mediate the resistance of alfalfa to aphids, namely AKR (*Acyrthosiphon kondoi* resistance), AIN (*Acyrthosiphon*-induced necrosis), APR (*Acyrthosiphon pisum* resistance), and TTR (*Therioaphis trifolii* resistance) [32,33,83,84,85]. Furthermore, several QTLs that regulate *M. truncatula* resistance or tolerance to aphids have been reported [32,86,87]. AKR and AIN mediate the resistance of *M. truncatula* to the blue-green aphid (*Acyrthosiphon kondoi* Shinji, BGA). AKR is a dominant resistance gene that plays a resistance role in the phloem and is involved in antixenosis and inducible antibiosis [83]. AIN controls the allergic necrotic response of *M. truncatula* to blue-green and pea aphids and significantly reduces the feeding on alfalfa of both aphids [32,33]. TTR mediates *M. truncatula* resistance to alfalfa-spotted aphids, while APR, localized in the NLR gene cluster on chromosome 3, mediates *M. truncatula* resistance to pea aphids [34]. Studies on plant resistance to piercing-sucking mouthpart pests suggest that most of the genes affecting resistance are NLR genes. For example, plant resistance to potato aphids is mediated by the NLR-like gene *Mi1.2* [88], and resistance to cotton aphids is mediated by the CC-type NLR gene *Vat* [89]. In terms of downstream signaling in *M. truncatula* insect resistance, it was identified that the signaling of the insect resistance response is dependent on phytohormone signals such as SA, JA, and ET [90]. For example, the expression of JA synthesis-related genes was significantly upregulated in resistant cultivars upon aphid infestation [91].

In addition to the resistance genes mentioned above, the gene encoding the insect protease inhibitor was transferred into alfalfa, and the transgenic plants exhibited a significant poisoning effect on thrip [36]. Bt crystal protein is an insecticidal insoluble crystal protein produced by *Bacillus thuringiensis* during spore germination. The researchers modified 249 of its 630 amino acids in the wild-type *Bt* gene (*cryIC*) to make it suitable for expression in plants, then artificially synthesized the *cryIC* gene and transferred it into alfalfa. Cry IC toxin content in transgenic alfalfa leaves ranged from 0.01 to 0.10 percent of total soluble protein. The leaves of transgenic plants were fed with larvae of the Egyptian sea gray leaf moth and the beet leaf moth, and the mortality rate of larvae was as high as 100% [35].

## 5. Problems and Future Directions of Research on Alfalfa Resistance to Pests and Diseases

### 5.1. Research Problems in Alfalfa Resistance to Pests and Diseases in China

In recent years, Chinese scholars have made considerable progress in the research of alfalfa pests and diseases, but the majority of their efforts have focused on the identification of pathogens in the field, the resistance evaluation of alfalfa varieties, and the screening of chemical agents. Research on the genetic basis and breeding of alfalfa resistance to disease and insect pests is still at the primary level. It is critical to establish an efficient and accurate alfalfa disease and pest resistance evaluation technology platform to systematically screen and identify pest- and disease-resistant germplasm resources. On the other hand, the research on the mechanism of alfalfa resistance to pests and diseases mostly focuses on the model species *M. truncatula*. There are few studies on the exploitation of resistance gene resources of alfalfa germplasms that are widely used, and durable broad-spectrum resistance gene resources are limited. This is mainly limited by the characteristics of alfalfa, such as cross-pollination, polyploidy inheritance, and self-incompatibility, which makes it difficult to mine molecular markers associated with excellent traits, which hinders the molecular breeding of alfalfa. Meanwhile, the long breeding cycle, imperfect breeding system, and insufficient capital investment are also important factors restricting alfalfa resistance breeding in China.

### 5.2. The Future Development Direction of Alfalfa Resistance Research

Based on the current research status of alfalfa resistance to disease and insect pests in China, the author believes that the strategic layout can be described by the following aspects in the future:

#### 5.2.1. Using Genome-wide Association Studies to Explore Alfalfa Resistance Genes

Chinese scholars have currently obtained the genomes of the alfalfa varieties Xinjiang Daye and Zhongmu No. 1 [92,93]. The further use of genome-wide association studies (GWAS) to explore the excellent gene resources of alfalfa resistance to disease and insects, and based on synthetic biology and gene editing technology, will result in fast molecular breeding of alfalfa that is resistant to pathogens and insects. For example, using GWAS technology, the maize banded leaf and sheath blight resistance gene ZmFBL41 was successfully isolated and identified [94]. Four wheat rust resistance genes were identified quickly using a combination of association genetics and R-gene enrichment sequencing (AgRenSeq) technology [95].

#### 5.2.2. Modification of Alfalfa’s Susceptible Genes by Gene-Editing Technology

Genes in plants that can promote pathogen infection can be called susceptibility (S) genes [96]. Pathogens exploit S genes to facilitate their colonization. The concept of the S gene is of great significance for breeding resistant plants. Disrupting these S genes may interfere with the compatibility between pathogens and host, and consequently, provide broad-spectrum and durable disease resistance. For example, *Mildew resistance locus O* (MLO) is a well-characterized S gene in barley [97]. The mutation of *MLO* confers durable and broad-spectrum resistance to powdery mildew in various plant species [97,98]. A new wheat germplasm harboring the Tamlo-R32 mutant allele was obtained using CRISPR-Cas9 technology, which conferred robust disease resistance without undesirable pleiotropic effects [99]. Rice’s resistance to the brown planthopper can be improved by mutating or silencing the cytochrome P450 gene *CYP71A1*, an S gene that can synthesize serotonin in rice [100]. These studies suggest that modifying S genes will also contribute to the improvement of alfalfa resistance.

#### 5.2.3. Using Pathogen Effectors as Probes to Explore the Immune Mechanisms of Alfalfa

During infection, pathogens and insects secrete effector proteins into or between plant cells to interfere with the plant immune system during infection. Focusing on the important pathogens or insects of alfalfa, identifying the effector proteins secreted during infection, and analyzing the molecular mechanisms of the effector proteins triggering or inhibiting host immunity will provide theoretical guidance for the application of the resistant genes. For example, PsXEG1 is an apoplastic effector secreted by a severe soybean pathogen *P. sojae* [47]. PsXEG1 not only promotes pathogenicity through its glycoside hydrolase activity but also could be recognized as a PAMP-like effector protein by plants [47]. A further study identified an RLP receptor NbRXEG1 from tobacco that can recognize PsXEG1, which is of great value for broad-spectrum resistance improvement in plants [46].

#### 5.2.4. Establish a Theoretical and Technical System of Alfalfa-Microbe/Microbiome Interaction, and Explore Alfalfa’s Microbial Resources for Pest and Disease Resistance

The interaction between *Arabidopsis thaliana* and its microbiome has opened up a new field of plant–microbiome interaction, and it has been found that the plant rhizosphere microbiome is closely related to plant immunity [101,102]. Studies on the triple interaction of pathogens, beneficial microorganisms, and plants should be strengthened to clarify the differences in the molecular response mechanisms of how plants seek advantages and avoid disadvantages, as well as explore beneficial microbial resources. Using a beneficial endophytic fungus, Chinese scholars have made significant advances in forage breeding. A new germplasm of barley with endophytic fungus *Epichloë bromicola* was generated by artificially inoculated *E. bromicola* isolated from wild barley to the cultivated barley variety, which showed higher biomass and seed yield [103]. These results indicated that conducting alfalfa microbiome research and mining beneficial microbial resources has important theoretical value and application potential for alfalfa resistance research.

#### 5.2.5. Application of New Technologies to Control Alfalfa Diseases and Pests

Many new pest and disease control technologies have been developed, providing more options for the control of alfalfa diseases and pests. For example, based on the theory of plant immunity, the development of immune inducers that can activate plant immunity can improve the resistance of plants to pests and diseases in a green and environmentally friendly manner [104,105]. Small RNAs play an important role in the resistance to fungi, oomycetes, and animal pathogens, and small RNAs can be transported across the host–pathogen boundaries. Plant small RNAs are novel “weapons” for the plant to defend against pathogens. For example, the application of host-induced gene silencing technology (HIGS) to silence the virulence gene *VdH1* of *V. dahliae* in cotton significantly enhanced the disease resistance of cotton to *V. dahliae* [106].

In conclusion, advances in genome sequencing and gene editing will enable us to explore the enormous arsenal of resistance genes hidden in alfalfa and their potential for resistance engineering. Moreover, breeding resistant cultivars with effective resistance genes, together with other novel plant protection technologies, will greatly improve alfalfa production in the future.

## Figures and Tables

**Table 1 plants-11-02008-t001:** Disease or insect resistance genes or QTL identified in alfalfa or *Medicago truncatula*.

Diseases or Pests	Pathogens or Insect	Gene(s) or QTL	Phenotype	Donor Species	Recipient Plants	References
Bacterial wilt	*Rastonia solanacearum*	*MtQRRS1*	Resistant	*Medicago truncatula*	*M. truncatula*	[7,8]
Bacterial wilt	*R. solanacearum*	*AtEFR*	Resistant	Arabidopsis	*M. truncatula*	[9]
Bacterial stem blight	*Pseudomonas syringae*	Lactoferrin-encoding gene	Resistant	Human	*M. sativa*	[10]
*Phytophthora* root rot	*Phytophthora medicaginis*	*MtERF1-1*	Resistant	*M. truncatula*	*M. truncatula*	[11]
*Phytophthora* root rot	*P. medicaginis*	Fungal β-1,3-glucanase	Resistant	*Trichoderma harzianum*	*M. sativa*	[12]
*Phytophthora* root rot	*P. palmivora*	*MtRAM2*	Susceptible	*M. truncatula*	*M. truncatula*	[13]
*Phytophthora* root rot	*P. palmivora*	*MtRAD1*	Susceptible	*M. truncatula*	*M. truncatula*	[14]
Oomycete root rot	*Aphanomyces euteiches*	F-box protein-encoding gene	Resistant	*M. truncatula*	*M. truncatula*	[15]
Oomycete root rot	*A. euteiches*	Chalcone O-methyltransferase	Resistant	*M. truncatula*	*M. truncatula*	[16]
Oomycete root rot	*A. euteiches*	*MtTPS10*	Resistant	*M. truncatula*	*M. truncatula*	[17]
Oomycete root rot	*A. euteiches*	*MtLYK9*	Resistant	*M. truncatula*	*M. truncatula*	[18]
Oomycete root rot	*A. euteiches*	*MtNF-YA1*	Susceptible	*M. truncatula*	*M. truncatula*	[19]
Oomycete root rot	*A. euteiches*	*AER1*	Resistant	*M. truncatula*	*M. truncatula*	[20]
Downy mildew	*Hyaloperonospora arabidopsidis*	*MtDef4*	Resistant	*M. truncatula*	Arabidopsis	[21]
Anthracnose	*Colletotrichum trifolii*	*RCT1*	Resistant	*M. truncatula*	*M. sativa*	[22,23]
Anthracnose	*C. trifolii*	*PALM1*	Susceptible	*M. truncatula*	*M. truncatula*	[24]
Powdery mildew	*Erysiphe pisi*	*MtREP1*	Resistant	*M. truncatula*	*M. truncatula*	[25]
*Verticillium* wilt	*Verticllium alfalfae*	*MsVR38*	Susceptible	*M. sativa*	*M. sativa*	[26]
*Verticillium* wilt	*V. alfalfae*	*MsVR39*	Resistant	*M. sativa*	*M. sativa*	[26]
*Verticillium* wilt	*V. alfalfae*	qVW-8C	Resistant	*M. sativa*	*M. sativa*	[27]
Fungal leaf disease	*Phoma medicaginis*	Isoflavone-O-transmethylase	Resistant	*M. sativa*	*M. sativa*	[28]
Rust	*Puccinia triticina*	*MtDef4*	Resistant	*M. truncatula*	Wheat	[29]
*Fusarium* root rot	*Fusarium oxysporum*	*MtABCG10*	Resistant	*M. truncatula*	*M. truncatula*	[30]
*Rhizoctonia* root rot	*Rhizoctonia solani*	Isoflavone synthase	Resistant	*M. truncatula*	*M. truncatula*	[31]
Blue-green aphid	*Acyrthosiphon kondoi*	*AKR*	Resistant	*M. truncatula*	*M. truncatula*	[32]
Blue-green aphid	*A. kondoi*	*AIN*	Resistant	*M. truncatula*	*M. truncatula*	[33]
Spotted aphid	*Therioaphis trifolii*	*TTR*	Resistant	*M. truncatula*	*M. truncatula*	[34]
Pea aphid	*A. pisum*	*APR*	Resistant	*M. truncatula*	*M. truncatula*	[34]
Beet leaf moth	*Scrobipalpa ocellatella*	Bt	Resistant	*Bacillus thuringiensis*	*M. sativa*	[35]
Thrip	Thrip	Proteinase inhibitor gene	Resistant	*Manduca sexta*	*M. sativa*	[36]

## Data Availability

Data sharing not applicable.

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
