# Peer review of "Research Progress and Prospect of Alfalfa Resistance to Pathogens and Pests"

_plants, 2022, doi:10.3390/plants11152008_

Round 1
Reviewer 1 Report
This study is aimed to review alfalfa disease and pest resistance. The study tries to give some interesting points on the identification of resistance genes against pests and diseases and related molecular mechanisms in alfalfa. The study is missing several aspects (e.g. no illustration, tables and figures) that the authors need to add before the study can be considered for possible publication.
Comments and suggestions
Abstract: The abstract is too general. The last two sentences are the aims of the review. You should give more info. You should make a last sentence conclusion at the end of the abstract.
L20: Before this section you need to add: ’Introduction’
L20-24: This section needs citations.
L25-31. Many general sentences but all are built on previous knowledge, so again you need to give citations.
L31-33: Here you said ’ Significant advances in plant immunity have been made over the years, and the framework of plant immunity theory has been well established’ – but no proof of citations.
L61-130: You need to create a table for summarizing features of receptors that you mentioned in the text.
L142-243: A summary table needs for resistance-related genes to bacteria, oomycete, and fungi
L224-243: This section (2.4) should be 2.1. as bacteria are taxonomically before oomycete.
L244-247: Pest section is quite short. A summary table needs for resistance-related genes to pests.
Author Response
Reviewer 1
Comments and Suggestions for Authors
This study is aimed to review alfalfa disease and pest resistance. The study tries to give some interesting points on the identification of resistance genes against pests and diseases and related molecular mechanisms in alfalfa. The study is missing several aspects (e.g. no illustration, tables and figures) that the authors need to add before the study can be considered for possible publication.
Comments and suggestions
Abstract: The abstract is too general. The last two sentences are the aims of the review. You should give more info. You should make a last sentence conclusion at the end of the abstract.
Response: We really appreciate the reviewer for this suggestion. The abstract has been revised, and the conclusion sentence has been added as “Breeding resistant cultivars with effective resistance genes, together with other novel plant protection technologies will greatly improve alfalfa production”.
L20: Before this section you need to add: ’Introduction’
Response: corrected.
L20-24: This section needs citations.
Response: The references have been added to the revised manuscript.
L25-31. Many general sentences but all are built on previous knowledge, so again you need to give citations.
Response: The references have been added to the revised manuscript.
L31-33: Here you said’ Significant advances in plant immunity have been made over the years, and the framework of plant immunity theory has been well established’ – but no proof of citations.
Response: The references have been added to the revised manuscript.
L61-130: You need to create a table for summarizing features of receptors that you mentioned in the text.
Response: We really appreciate the reviewer for this suggestion. The topic of this paper is related to the researches on alfalfa resistance to diseases and pests. The description of the plant immunity theory of other plants is to lay the foundation for the introduction of the researches on alfalfa, because there are few researches on the plant immunity theory on alfalfa. On the other hand, we only list some typical plant immune receptor studies as examples, since there are several excellent and comprehensive reviews for the summary of plant immune receptors (Boutrot et al., 2017; Kapos et al., 2019). Therefore, we have not summarized immune receptors from other plants in this paper.
References
Boutrot, F., Zipfel, C. Function, discovery, and exploitation of plant pattern recognition receptors for broad-spectrum disease resistance. Annu Rev Phytopathol, 2017. 55: p. 257-286.
Kapos, P., Devendrakumar, K.T., Li, X. Plant NLRs: From discovery to application. Plant Sci, 2019. 279: p. 3-18.
L142-243: A summary table needs for resistance-related genes to bacteria, oomycete, and fungi
Response: Thanks for this suggestion. Resistance-related genes to bacteria, oomycete, fungi, and pests have been summarized in Table 1 in the revised manuscript. We have gone through the literatures and added some new citations in Table 1.
L224-243: This section (2.4) should be 2.1. as bacteria are taxonomically before oomycete.
Response: The bacteria section (2.4) has been moved to the section before oomycete in the revised manuscript.
L244-247: Pest section is quite short. A summary table needs for resistance-related genes to pests.
Response: Thanks, we agree with the reviewer's point of view that this part is really short. In fact, there are few studies on insect resistance genes of alfalfa, and most of genes comes from M. truncatula. Resistance-related genes to pests have been summarized in Table 1 in the revised manuscript.
Reviewer 2 Report
Dear Authors,
This is a rigorous scientific effort. I made comments and suggestions throughout the text. Please go through those carefully and improve those. Particularly consider improving the introduction, objectives, and conclusion part.
Here are some suggestions:
The title is common and not catchy.
In the abstract and introduction need to add a little about the uses and nutritional value of alfalfa as a forage crop and why it is so important.
Introductions need to have the title with little more elaboration, specially on the previous research, current research, future directions, and clear objectives.
There are headings and under which lots of sub-headings with numbers, that confused me. Please organized the headings and sub-headings and it will be better if you don’t use numerical orders.
Please add the scientific name of Alfalfa, Line 20.
Please rewrite the sentence, check article and conjunction and make it meaningful and complete. Line 20-21
Where is reference here? Also while citing some researchers please add name with year of publication. Line 39-40
Please write the downstream defense event sequentially instead of just listing it. Line 115-117
Why passive here, rewrite it. Line 134
Rewrite it, sentence is not clear. Line 146
Please rewrite and make the title shorter and simple, also its "fungal diseases”. Line 172
Line 213, delimit the space between to and Verticillium
How is this line connected with rest of the paragraph? Line 220-222
Line 246-248. Please provide some statistics of Alfalfa insect damage related loss with reference from China. When you are using US statistics, please add reference from the USDA statistics. Line246-248
Line 284, you may say still in the primary level instead of infancy as it is not an appropriate word here.
293-295, rewrite the sentence to make it simple and clear.
You should write a conclusion under the heading with future research prospects and directions to make the review a complete one.
Thank you.

Author Response
Reviewer 2
Dear Authors,
This is a rigorous scientific effort. I made comments and suggestions throughout the text. Please go through those carefully and improve those. Particularly consider improving the introduction, objectives, and conclusion part.
Here are some suggestions:
The title is common and not catchy.
Response: Thanks, the title has been revised as “Research progress and prospect of alfalfa resistance to pathogens and pests”.
In the abstract and introduction need to add a little about the uses and nutritional value of alfalfa as a forage crop and why it is so important.
Response: The importance of alfalfa has been added to the revised manuscript in abstract as ‘Alfalfa is one of the most important legume forages in the world and contributes greatly to the improvement of ecosystems, nutrition and food security.’ and in introduction as ‘Alfalfa (Medicago sativa) is the world's most significant perennial legume forages, with large yields, high digestibility, good palatability, and wide adaptability. Alfalfa is regarded as "the queen of forages" as it contributes greatly to the improvement of eco-systems, nutrition and food security’.
Introductions need to have the title with little more elaboration, specially on the previous research, current research, future directions, and clear objectives.
There are headings and under which lots of sub-headings with numbers, that confused me. Please organized the headings and sub-headings and it will be better if you don’t use numerical orders.
Response: Thank you. The introduction title has been added and defined as the first section.
Sorry for the confusion, we organized the headings and sub-headings with numerical orders according to the format requirement of Plant journal.
Please add the scientific name of Alfalfa, Line 20.
Response: Corrected.
Please rewrite the sentence, check article and conjunction and make it meaningful and complete. Line 20-21
Response: The sentence has been revised as ‘Alfalfa (Medicago sativa) is the world's most significant perennial legume forages, with large yields, high digestibility, good palatability, and wide adaptability. Alfalfa is regarded as "the queen of forages" as it contributes greatly to the improvement of ecosystems, nutrition and food security.’
Where is reference here? Also while citing some researchers please add name with year of publication. Line 39-40
Response: The reference has been added to the revised manuscript.
Please write the downstream defense event sequentially instead of just listing it. Line 115-117
Response: Corrected. ‘This process includes the protein phosphorylation/ubiquitination regulation, synthesis of reactive oxygen species, activation of the MAPK signaling pathway, expression of defense response genes, synthesis of secondary metabolites, etc’.
Why passive here, rewrite it. Line 134
Response: Corrected as ‘A variety of pathogens infect alfalfa and cause serious diseases, thus affects yield and quality of alfalfa’.
Rewrite it, sentence is not clear. Line 146
Response: Corrected as ‘These soil-borne pathogens can infect alfalfa seedlings or mature plants especially in soil fields with high humidity or poor drainage’.
Please rewrite and make the title shorter and simple, also its "fungal diseases”. Line 172
Response: Corrected as ‘Fungal disease resistance genes and mechanisms in alfalfa’.
Line 213, delimit the space between to and Verticillium
Response: Corrected.
How is this line connected with rest of the paragraph? Line 220-222
Response: We used M. truncatula defensin-related gene MtDef4 as an example here is to demonstrate the disease resistance-related genes from M. truncatula could also work well in other plants to defense against fungal pathogens.
Line 246-248. Please provide some statistics of Alfalfa insect damage related loss with reference from China. When you are using US statistics, please add reference from the USDA statistics. Line246-248
Response: The statistics of Alfalfa insect damage related loss from China and related reference have been added to the revised manuscript as ‘Based on 2017 data, the annual loss of alfalfa in China caused by insect pests exceeds 140 million US dollars (McNeill et al., 2022)’.
Reference:
McNeill, M.R., Tu, X., Altermann, E., Beilei, W., Shi, S. Sustainable Management of Medicago sativa for Future Climates: Insect Pests, Endophytes and Multitrophic Interactions in a Complex Environment. Frontiers in Agronomy, 2022. 4.
Line 284, you may say still in the primary level instead of infancy as it is not an appropriate word here.
Response: Corrected.
293-295, rewrite the sentence to make it simple and clear.
Response: this sentence has been revised as ‘Meanwhile, long breeding cycle, imperfect breeding system and insufficient capital investment are also important factors restricting alfalfa resistance breeding in China’.
You should write a conclusion under the heading with future research prospects and directions to make the review a complete one.
Response: Thank you. The conclusion has been added to the revised manuscript as ‘In conclusion, advances in genome sequencing and gene editing will enable us to explore the enormous arsenal of resistance genes hidden in alfalfa and their potential for resistance engineering. Moreover, breeding resistant cultivars with effective resistance genes, together with other novel plant protection technologies will greatly improve al-falfa production in the future’.
Reviewer 3 Report
Dear authors and editor,
the manuscript "Research progress and prospect of alfalfa disease and pest resistance" presents a review in the advance of alfalfa resistance research. The topic is interesting for the scientific community as it provides an overview of the research done so far. The manuscipt is very concise and the first part sounds more like a book chapter than a review. Furthermore, too many statements don't report a reference (Lines 21-23-40-42-45-69-75-88-98-130-179-207-208-210-211-248-331). This must be amended before accepting the manuscript. The second part is far more interesting as it focuses on the future direction of research.
You should improve the first section and add a short conclusion.
Author Response
Reviewer 3
Comments and Suggestions for Authors
Dear authors and editor,
the manuscript "Research progress and prospect of alfalfa disease and pest resistance" presents a review in the advance of alfalfa resistance research. The topic is interesting for the scientific community as it provides an overview of the research done so far. The manuscipt is very concise and the first part sounds more like a book chapter than a review. Furthermore, too many statements don't report a reference (Lines 21-23-40-42-45-69-75-88-98-130-179-207-208-210-211-248-331). This must be amended before accepting the manuscript. The second part is far more interesting as it focuses on the future direction of research.
Response: Thanks for the suggestion, the references (Lines 21-23-40-42-45-69-75-88-98-130-179-207-208-210-211-248-331) have been added to the revised manuscript.
You should improve the first section and add a short conclusion.
Response: We really appreciate the reviewer for this suggestion. The topic of this paper is related to the researches on alfalfa resistance to diseases and pests. The description of the plant immunity theory of other plants is to lay the foundation for the introduction of the researches on alfalfa, because there are few researches on the plant immunity theory on alfalfa. On the other hand, in order to highlight the emphasis of this review on alfalfa resistance to disease and insect, we did not introduce plant immunity research in detail, but only described the concept of plant immunity through some examples. We are sure that readers will benefit from a more detailed introduction to plant immunity, and several excellent and comprehensive reviews for the summary of plant immune receptors (Boutrot et al., 2017; Kapos et al., 2019) have been added to the revised manuscript.
The short conclusion of the first section has been added to the revised manuscript as ‘Annually, about twenty percent of crop yields are lost due to plant diseases and pests, which affect food security worldwide (Oerke et al., 2006). Pathogens constantly evolve to overcome host resistance. In addition to traditional chemical control strategies, improved understanding of plant immunity should contribute to the development of new crop protection strategies. Genetic methods to improve crop resistance with plant immune receptors will provide broad-spectrum disease resistance. Although studies on plant immunity are mostly reported on model plants and staple crops, it will be helpful to identify immune receptors and immune-related genes of alfalfa through these studies to enable new breeding strategies to enhance alfalfa disease resistance’.
References
Boutrot, F., Zipfel, C. Function, discovery, and exploitation of plant pattern recognition receptors for broad-spectrum disease resistance. Annu Rev Phytopathol, 2017. 55: p. 257-286.
Kapos, P., Devendrakumar, K.T., Li, X. Plant NLRs: From discovery to application. Plant Sci, 2019. 279: p. 3-18.
Oerke, E.C. Crop losses to pests. The Journal of Agricultural Science, 2005. 144: p. 31-43.
Round 2
Reviewer 2 Report
Dear Author,
Thank you for editing as suggested. Also I liked the table 1 that you added. Thank you for improving the content of the review.
Author Response
Dear reviewer:
Thanks for your thoughtful and detailed input.
Reviewer 3 Report
Dear authors, you did a great effort to improve the manuscript quality. I still think the first part of the manuscript seems more a book chapter but it is compensated by the quality of the second part. I think the manuscript may be considered for publication.Author Response
Dear reviewer:
Thanks for your thoughtful and detailed input. We are grateful for the chance to strengthen our manuscript in light of your comments. The language of the manuscript has been revised according to the suggestions of professional editor of Plants.